# A Biochemical and Histological Assessment of Postmortem Changes to the Eyes of Domestic Pigs: A Preliminary Study

**DOI:** 10.3390/ani14081190

**Published:** 2024-04-15

**Authors:** Magdalena Palić, Ivan-Conrado Šoštarić Zuckermann, Petar Džaja, Blanka Beer Ljubić, Krešimir Severin

**Affiliations:** 1Department of Forensic and State Veterinary Medicine, Faculty of Veterinary Medicine, University of Zagreb, Heinzelova 55, 10000 Zagreb, Croatia; mpalic@vef.unizg.hr (M.P.); dzaja@vef.unizg.hr (P.D.); 2Department of Veterinary Pathology, Faculty of Veterinary Medicine, University of Zagreb, Heinzelova 55, 10000 Zagreb, Croatia; isostaric@vef.unizg.hr; 3Internal Diseases Clinic, Faculty of Veterinary Medicine, University of Zagreb, Heinzelova 55, 10000 Zagreb, Croatia; bljubic@vef.unizg.hr

**Keywords:** domestic pigs, biochemistry, electrolytes, vitreous of the eye, histology, retina

## Abstract

**Simple Summary:**

From a veterinary forensic point of view, an accurate estimation of the Postmortem Interval (PMI) is important when considering the worldwide increase in deaths of domestic and wild animals. A preliminary study was conducted using the eyes of domestic pigs. A biochemical analysis was conducted on the vitreous humor of the eye, whilst a histological analysis was conducted on the retina of the eye. The eyes were stored at +4 °C and changes were monitored at time intervals of 0, 12, 24, 48, and 120 h.

**Abstract:**

The Postmortem Interval (PMI) is the time from the death of an animal to its discovery. From a veterinary forensic standpoint, an accurate estimation of the PMI is of particular importance, especially with the observed increase in deaths of domestic and wild animals. A preliminary study was conducted using the eyes of domestic pigs. A biochemical analysis was conducted on the vitreous humor of the eye, whilst a histological analysis was conducted on the retina. The eyes were stored at +4 °C and changes were assessed at time intervals of 0, 12, 24, 48, and 120 h. The biochemical analysis during the PMI established a decrease in sodium, chlorine, and glucose concentrations, and a rise in potassium concentration. Accordingly, a simple linear regression showed a significant correlation between changes in concentrations of sodium (Na^+^), potassium (K^+^), chloride (Cl^−^), and glucose, in relation to the PMI. The histological analysis showed evident morphological changes in the retina, which included homogenization of the rod and cone cells, pyknosis of the outer nuclear layer, homogenization of the outer plexiform layer, pyknosis of the inner nuclear layer, homogenization of the inner plexiform layer, and pyknosis of the nuclei of the ganglion layer of the retina.

## 1. Introduction

The Postmortem Interval (PMI) is the time from the death of an animal to its discovery. Estimating the PMI is an important aspect of forensic investigations, therefore particular attention should be paid to practical as well as scientific research. There is less research in this regard in veterinary forensics than in human forensics, but the need for precise assessment of the PMI is increasing. Death, regardless of how it occurs, represents the end of life functions, which includes the cessation of circulation. As a result, acute hypoxic–ischemic injuries occur to cells and tissue [1,2], along with changes to their morphology and function. Depending on the type of cell or tissue, and the intensity and type of their metabolism, these changes take place in a specific order. Therefore, we can say that death is not a single moment, but a process that continues over a specific period of time. In forensic pathology, these changes have particular importance for the assessment of the PMI [3]. The methods used for this assessment are divided into subjective or quantitative methods [4]. Until now, the PMI has mainly been established on the basis of a subjective assessment of the macroscopic postmortem changes in the body, and forensic entomology. However, it should be pointed out that their intensity/occurrence changes under the influence of a large number of internal and external factors. Therefore, the establishment of the PMI on the basis of these changes alone is extremely imprecise, that is, it is only possible in the early phases after death [5]. As a result, in both human and veterinary forensics, efforts are being made to increase the precision of these estimates, and many methods are being researched and validated. Postmortem biochemistry (*thanatochemistry*) is one of the methods that has become an important aspect of forensic investigations in recent years [6,7]. Most biochemical analyses are based on determining changes in the concentrations of electrolytes, products of the metabolism, products of the metabolism of bacteria, and the decomposition of proteins [4]. Biochemical research has been conducted on bodily fluids such as blood [8], cerebrospinal fluid, the aqueous humor [8], synovial fluid [9,10], and the vitreous humor of the eye. Due to its anatomic location and structure, its large volume, and the weaker reflection of changes in the blood [6,11], the vitreous humor has been shown to be an excellent substrate in terms of the assessment of the PMI. The presence of the blood–ocular barrier and its limited vascularization mean the vitreous humor is protected from the effects of external factors and bacterial contamination, therefore the final phase of postmortem decomposition is delayed for a longer period of time [10,11]. It should be pointed out that the vitreous humor remains preserved even in cases of cranial injury [7]. Research to date has mainly been conducted in relation to humans [12,13,14,15,16,17], and less to animals such as horses [18], dogs [19,20], cats [20,21], donkeys [10], pigs [22,23,24], sheep [25], cattle [20,24,25], rabbits [26], and red deer [27]. In modern forensics, histological findings, despite being a subjective method, play a crucial role and serve as a valuable complement to other investigative techniques. A very small amount of research has been conducted in relation to the assessment of the PMI using histological analysis, especially on eye tissue. Therefore, a histological analysis and the establishment of morphological changes to the retina would be of utmost significance.

The aim of this study was to describe the correlation between changes in the concentrations of biochemical parameters (Na^+^, K^+^, Cl^−^, and glucose) in the vitreous humor of the eye and morphological changes in the structure of the retina over time, after death (PMI).

## 2. Materials and Methods

### 2.1. Animal Sampling

Samples for conducting the preliminary study were collected from a total of fifteen domestic pigs (*Sus scrofa domesticus*) (*n* = 15) during the regular slaughter process at the Facility for Slaughter and Processing of Animals, Bebrinka d.o.o, Donja Bebrina, Ulica starog Hrasta 3, 35208, Donja Bebrina, Croatia. Further processing of the samples was conducted at the Department of Forensic and State Veterinary Medicine of the Veterinary Faculty of the University of Zagreb, and the Histology and Immunohistochemistry Laboratory within the Pathology Department of the Medical Faculty of the University of Zagreb.

### 2.2. Tissue Sampling

Immediately after slaughter, the eyeballs were enucleated. One eye was used for biochemical analysis and the other for histological analysis. The tissue was stored at +4 °C until further processing. During the sampling of the vitreous humor, five eyes incurred damage, leading to their exclusion from the samples.

### 2.3. Blood Sampling

At the moment of slaughter, as the jugular veins were cut, blood samples were taken in 5 mL Z SEP GEL SERUM test tubes, after which the blood was centrifuged at 3000 rpm ×10 min in order to separate the serum. The samples of serum were stored at −20 °C until processing. Before analysis, the samples were thawed and analyzed in an automatic chemical analyzer—Architect C4000 (Abbott Park, IL, USA). Electrolyte concentrations (Na^+^, K^+^, Cl^−^) and glucose levels (mmol/L) were established.

### 2.4. Sampling the Vitreous Humor

After sampling, the eyes were wrapped in paraffin and a needle was inserted permanently into the eye and fixed (20G, 0.9 mm × 38 mm) along the lateral edge of the optic nerve. The vitreous humor of the eye was sampled from a total of ten domestic pigs (N = 10), since, as already mentioned, in the case of five pigs the eyes were damaged during processing. The eyes for biochemical testing were stored at +4 °C and, using a vacuum system for collecting blood at each time interval (0, 12, 24, 48, and 120 h), 200 µL of vitreous humor was collected. The centrifuged samples of vitreous humor were stored at −20 °C until analysis. Before analysis, the samples were thawed and analyzed in an automatic chemical analyzer—Architect C4000 (Abbott Park, IL, USA). The concentrations of sodium (Na^+^), potassium (K^+^), chlorine (Cl^−^), and glucose (mmol/L) were measured.

### 2.5. Fixation and Histological Specimen Preparation—Categorical Analysis 

The eyes for the histological analysis were stored at +4 °C and fixed at time intervals of 0, 12, 24, 48, and 120 h. At each time interval, three eyes were fixed (N = 3). Davidson’s fixative was used for fixation for a period of 48 h. Routinely prepared histological samples (H&E staining) were examined using a Digicyte DX50 light microscope (Digicyte digitalne tehnologije d.o.o., Zagreb, Croatia). Images were captured with a Digicyte BigEye microscope camera (400, 600×). The histological analysis was based on a subjective assessment of the changes in the structure of the retina by two pathologists (independent and blinded assessments) according to the following given criteria: the assessment was conducted in ten randomly selected high-power fields, ensuring the presence of all retinal layers and minimal artifacts resulting from tissue trimming. Several criteria were assessed, which included detachment of the neuroretina from the retinal pigment epithelium, pyknosis of the outer nuclear layer, pyknosis of the inner nuclear layer, pyknosis of the ganglion cell layer, homogenization of the cones and rods cells, homogenization of the outer plexiform layer, and homogenization of the inner plexiform layer. The recorded changes were graded at four levels: Absent (0%) −, Mild (1–30%) +, Moderate (31–70%) ++, and Severe (71–100%) +++.

### 2.6. Statistical Data Analysis 

The statistical analysis of data was conducted using the Statistica computer program (TIBCO Statistica^®^ 14.1.0). Descriptive statistics were summarized for all the parameters tested, which included the determination of central tendency and variation. The Kolmogorov–Smirnov test was used to test the normality of the distribution of the studied variables. The results of the histological analysis were defined as the median of ten visual fields per criterion. In order to determine the significance of the relationships between the concentrations of the biochemical parameters and the histological changes in the structure of the retinas over the course of time from death (the PMI), simple linear regression and Spearman’s rank correlation tests were performed. The differences between the groups were tested using a one-way ANOVA. The significance of the differences between the concentrations of the biochemical parameters in the vitreous of the eyes and their concentrations in serum was determined using the Student’s *t*-test method and simple linear regression. The statistical hypotheses were tested to a level of significance of *p* < 0.05.

## 3. Results

### 3.1. The Results of the Biochemical Analysis of Serum

The mean values of biochemical parameters in the serum were as follows: sodium 146.40 mmol/L, potassium 7.29 mmol/L, chloride 105.40 mmol/L, and glucose 5.74 mmol/L, while the mean values of the vitreous humor concentrations were as follows: sodium 143.70 mmol/L, potassium 5.76 mmol/L, chloride 123.80 mmol/L, and glucose 2.20 mmol/L. The results of the Student’s *t*-test showed that there was a significant difference in the concentrations of the biochemical parameters between the serum and the vitreous humor, sodium (*p* = 0.028), potassium (*p* = 0.000), chloride (*p* = 0.000), and glucose (*p* = 0.000) (Table 1).

### 3.2. The results of the Biochemical Analysis of the Vitreous Humor of the Eye

For all samples and all postmortem intervals, the concentrations of sodium (Na^+^) ranged in value from 146 mmol/L to 124 mmol/L. The simple linear regression method showed that the correlation between sodium concentration and the postmortem interval (PMI) was statistically significant, with a significant fall in relation to the passing of time after death up until 120 h (R2 = 0.750, *p* = 0.000) (Figure 1).

The concentrations of potassium (K^+^) ranged in value from 5.0 mmol/L to 23.6 mmol/L. The correlation of potassium concentrations to the postmortem interval (PMI) was shown to be statistically significant, with a significant rise up until 120 h (R2 = 0.819, *p* = 0.000) (Figure 2).

The concentrations of chloride (Cl^−^) ranged in value from 102 mmol/L to 127 mmol/L. The correlation of chloride concentrations to the postmortem interval (PMI) was shown to be statistically significant, with a significant fall up until 120 h (R2 = 0.874, *p* = 0.000) (Figure 3).

The concentrations of glucose ranged in value from 0.170 mmol/L to 2.700 mmol/L. The correlation of glucose concentrations to the postmortem interval (PMI) was shown to be statistically significant, with a significant fall up until 120 h (R2 = 0.367, *p* = 0.000) (Figure 4).

The one-way ANOVA test established a significant difference between the concentrations in the individual groups (0, 12, 24, 48, and 120 h).

### 3.3. The Histological Analysis of the Retina 

The simple linear regression established a significant correlation between retinal morphological changes and the PMI, which included homogenization of the cone and rod cells (multiple R2 = 0.414, *p* = 0.010), pyknosis of the outer nuclear layer (R2 = 0.748, *p* = 0.000), pyknosis of the inner nuclear layer (R2 = 0.224, *p* = 0.043), homogenization of the inner plexiform layer (multiple R2 = 0.599, *p* = 0.000), homogenization of the outer plexiform layer (R2 = 0.599, *p* = 0.000), and pyknosis the ganglion cell layer (R2 = 0.809, *p* = 0.000, and with the PMI). No significant correlation was found between the detachment of the nervous layer of the retina from the retinal pigment epithelium (RPE) with the PMI (multiple R2 = 0.0357, *p* = 0.500) (Table 2, Figure 1).

## 4. Discussion

The estimation of the postmortem interval is a key element of veterinary forensic investigations. Various body fluids and tissues are being extensively researched to enhance the accuracy of PMI estimations. The vitreous body (corpus vitreum) of the eye is an anatomic structure of the eye, similar to gel, and is located between the lens and the retina. The vitreous humor (*humor vitreous*) is an acellular part of the vitreous body, and is mainly composed of water (99%), salt (0.9%), protein, and polysaccharides (0.1%) [11,28]. For cells to be able to perform their functions, they must have certain concentrations available of oxygen, ions, amino acids, fat, and other substances. As a result, the homeostasis of electrolytes and glucose is regulated by precise mechanisms [29]. When death occurs, vital functions cease, therefore certain changes are expected in the concentrations of these parameters in the body, including in the vitreous humor. In the present study, the changes we found In the concentrations of electrolytes (Na^+^, K^+^, Cl^−^) are in line with previous studies in pigs [22,23,24], as well as in other species [10,18,27]. In a study conducted on donkeys, a slight increase in sodium and chloride levels was recorded in the early PMI [10], which was not observed in our study. According to a previous study conducted exclusively on humans, postmortem concentrations of glucose in the vitreous humor make it possible to determine the cause of death, or otherwise, they indicate previous life-long hyperglycemia, or confirm diabetes mellitus and diabetic ketoacidosis, as well as other causes of death, such as asphyxia, congestive heart failure, the agonal period, cerebral hemorrhage, electrocution, and hypothermia [11]. Some studies have been conducted to assess the PMI on the basis of glucose concentrations [10,27,28,30]. In the present study, the concentrations of glucose decreased significantly after death in all the animals. The mechanism of these changes in biochemical parameters can be explained in several ways. During the early PMI, changes in electrolyte concentrations are the result of autolytic changes (hypoxic–ischemic injury) and the loss of the integrity of the cell membrane. Moreover, after death, the production of energy and the active transport of electrolytes cease. Therefore, according to Fick’s law of diffusion, electrolytes begin to move passively from places with higher concentrations to places with lower concentrations [4,11]. The decrease in the concentration of glucose in the vitreous humor is the result of the mechanism of anaerobic glycolysis [10,30,31]. It must be mentioned here that the vitreous of the eye contains a very small number of cells, therefore changes in the concentrations of electrolytes and glucose relate to the disintegration and metabolism of the surrounding cell structures, including primarily the retina and the choroid [11,28]. With regard to the nervous system, it is important to highlight the potential for neuronal cell death due to glutamate excitotoxicity or glutamate-induced neurotoxicity. As a result of these mechanisms, higher concentrations of glutamates promote the entrance of sodium and then chloride into the cells, thereby changing their osmolarity. As a result of this, cell edema, osmotic shock, and finally complete lysis can occur [1]. In the later phases after death, changes in the concentration of electrolytes and glucose may be the result of the decomposition of other cell structures of the eye and bacterial contamination, which lead to glycolysis [22,31]. In the present study, we aimed to reduce contamination to the lowest possible level. After enucleation, the eyes were wrapped in paraffin and a needle was permanently fixed beside the optic nerve in order to avoid repeated insertion and the possibility of contamination. It is important to emphasize that the changes in the concentrations of these biochemical parameters are affected by various factors, such as age, gender, various pathological conditions, the cause of death (drowning), the death struggle, the environmental temperature, and scavenger activity [13,15,32,33,34,35,36,37]. The present study excluded the possible influence of extreme electrolyte and glucose values on the concentrations in the vitreous of the eye. Blood was taken immediately after the jugular veins were cut, and the serum was separated from it to compare the concentrations of the biochemical parameters in the vitreous of the eye with those in the serum. The present study found a significant difference in the concentrations of electrolytes and glucose between the serum and the vitreous of the eye. Moreover, the biochemical parameters in the serum were within the reference range (https://laboklin.com, accessed on 20 March 2023), except for potassium concentrations, as established in pigs by McLaughlin and McLaughlin [24], where the potassium concentration ranged from 7.5 ± 2.0 mmol/L. In light of this, potassium levels that are higher than the reference values could be explained by the perimortem stressor and lethal agony, which lead to changes in the blood pH including extracellular/metabolic changes [38]. When assessing the PMI on the basis of a biochemical analysis, there should be particular emphasis on how the samples are collected, the processing of the vitreous humor before analysis, and the analysis itself [13,15,33,34,35]. The statistical analysis of the results should also be considered. Using multiple regressive analyses, considering the concentrations of several different parameters, the reliability of the assessment of the PMI of 14 h increases, in comparison with when it is assessed on the basis of only one parameter (potassium), where the reliability of the assessment is 16.2 h [4].

The vitreous of the eye contains a very small number of cellular elements, therefore changes in the concentrations of electrolytes and glucose relate to changes in other cell structures in the eye, primarily the retina and the choroid [11,28]. In view of this, the present study monitored the morphological changes in the structure of the retina. In terms of embryonic development, the retina is part of the central nervous system, except for the retinal pigment epithelium. Seen histologically, the inner membrane of the eyeball (tunica interna bulbi, retina), is divided into the blind anterior part (pars ceca retinae), which does not contain receptors that are sensitive to light, and the posterior part, which contains receptors sensitive to light (pars optica retinae). The present study monitored changes to the optical part of the retina. Its main task is to convert light energy into chemical energy, and finally into electrical impulses, which pass along the optic nerve into the optical center of the brain. It is divided into the outer pigment layer (stratum pigmentosum retinae) and the inner neuron layer (stratum nervosum retinae). The neuron layer of the retina is multi-layered, and part of it is the diencephalon. It contains glial cells (Müller glial cells) and nerve cells (neurons) [39]. The retina has an extremely complex metabolism and high energy requirements [40] which are on a higher level than the brain [1]. This would indicate that postmortem hypoxia and ischemia very quickly lead to irreversible injuries to these cells and their death [41]. However, the nerve cells of the retina are more resistant to these injuries than brain cells. The histological analysis of the retina showed significant changes during the PMI, where a significant increase was found in the number of pyknotic nuclei of the outer and inner nuclear layer, and pyknosis of the ganglion cell layer. The changes recorded in this study correspond completely with the changes recorded by Finnie et al. [42] in rat retinas. Moreover, a significant correlation was found between the intensity of the homogenization of the cones and rods, and between the homogenization of the outer and inner plexiform layer and the PMI. Despite the high intensity of the metabolism and the energy requirements of the cones and rods, the present study found that changes were primarily seen in the inner retinal layers. Inside the inner nuclear layer, it could be seen that the cells did not all pass through the morphological changes with the same intensity, therefore we could conclude that glial cells have greater resistance to ischemia/hypoxia, which is in line with current knowledge. The neurons of the retina mainly depend on oxidative metabolism, in contrast to glial cells, which can adjust to anaerobic conditions [1]. Why the retina, despite its high metabolism, can withstand hypoxia/ischemia, and why the structures of the cones and rods are preserved for a longer period may be explained by the presence of the respiratory protein, neuroglobin [43,44]. Depending on the type of neuron, its concentration varies, and it is higher in the retina than in the brain [45]. Neuroglobin correlates with the concentrations of mitochondria and cytochrome c oxidase, whilst the precise function of neuroglobin is unclear [1,46]. However, it should be stressed that it has a greater role in neuroprotection in the case of ischemia than in the transport of oxygen [47]. The present study did not find any significant correlation between the detachment of the nervous layer of the retina from the retinal pigment epithelium in relation to the PMI. This change was recorded in a previous study, but most probably it was an artefact resulting from fixation.

This study involved certain limitations that warrant acknowledgment. One limitation was that we did not use the same eye for both the biochemical and histological assessments. However, we were encouraged by previous studies in both humans and dogs that failed to identify any notable differences in biochemical parameter concentrations between the left and right eyes [15,19,33]. Operating under the presumption of similar biochemical profiles in both eyes, one eye from each domestic pig was dedicated to biochemical testing, while the other was allocated for histological analysis. It is imperative to recognize the preliminary nature of this study, which involved a limited number of samples, aimed primarily at gauging the significance of observed changes. Despite the modest sample size, this study served the crucial purpose of establishing accurate methods for sampling and processing. Special attention was given to the correct fixation of eye tissue to ensure optimal tissue morphology and prevent postmortem alterations. Despite these limitations, this study contributes significantly to veterinary forensics, particularly in addressing the rising incidence of deaths among domestic and wild animals. Beyond its relevance to deaths, the assessment of the postmortem interval (PMI) has proved valuable during infectious disease outbreaks, such as African swine fever. Determining the source and duration of infection became paramount in controlling the spread of the disease. In essence, while acknowledging these limitations, this study will have a positive impact on advancing knowledge in veterinary forensics and public health.

## 5. Conclusions

The preliminary study conducted here established methods for collecting and processing samples, with an emphasis on the creation of histological preparations for the eye. The biochemical and histological analyses of the structures of the eye found significant changes in relation to the PMI. To validate these methods, the number of samples will be increased in future studies, whilst individual segments of the histological analysis will be developed further, with the use of state-of-the-art methods. Furthermore, future studies will investigate how changes in environmental temperature affect the biochemistry and histological changes of the eye. 

## Data Availability

The data presented in this study are available on request from the first author.

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
