# Peer review of "A Biochemical and Histological Assessment of Postmortem Changes to the Eyes of Domestic Pigs: A Preliminary Study"

_animals, 2024, doi:10.3390/ani14081190_

Round 1

Reviewer 1 Report

Comments and Suggestions for Authors

Overall this is a well-written paper discussing the use of biochemical analysis of vitreous humor and postmortem changes that occur in the eyes of pigs.  The findings provide good support for the continued exploration of this area of research.

Line 11 – you cannot have an exact estimation.  Remove “exact”

Line 11 and throughout the paper – Postmortem is one word.  No hyphen.  Change throughout.

Line 12 – Do the deaths have to be violent?  Suggest removing ‘violent’ reference as it can be used for all types of deaths.

Line 18 – “Exact – see above

Line 31 – Suggest adding “electrolytes” to the list

Line 34 – See postmortem comment above

Line 46 – Entomology should be listed here as there are a number of studies including on animals (pig is a human model)

Line 53 (and other locations) – “Post-mortal” change to “postmortem”

Lines 66-68 – While This is not an exhaustive list there are papers that include cats and bovine as well.  Suggest listing cats as they are one of the most frequently abused animal species.

·         Hanna PE, Bellamy JE, Donald A. Postmortem eyefluid analysis in dogs, cats and cattle as an estimate of antemortem serum chemistry profiles. Can J Vet Res. 1990 Oct;54(4):487-94. PMID: 2249181; PMCID: PMC1255698.

·         Stern AW, Roig D, Valerio C, Denagamage T. Postmortem Analysis of Vitreous Urea Nitrogen, Creatinine, and Magnesium of Renal and Post-Renal Disease in Cats. Toxics. 2023 Aug 10;11(8):685. doi: 10.3390/toxics11080685. PMID: 37624190; PMCID: PMC10458759.

·         McCoy MA. Hypomagnesaemia and new data on vitreous humour magnesium concentration as a post-mortem marker in ruminants. Magnes Res. 2004 Jun;17(2):137-45. PMID: 15319147.

Line 186 – Change “cones and rods cells” to “cone and rod cells”

Line 196 (Figure 1) –  The scales used in the inserts are too small to easily read.  Can you enlarge them?  The reference arrows are not in the figure.  Suggest changing the 40x and 60x reference to 400x and 600x as magnification is typically listed as the power of the objective (4X, 10X, 40x) and multiplied by the power of the eyepiece, usually 10X.

Line 211 – no need to bold “corpus vitreum”

Line 226 – add ‘mellitus’ after diabetes

Line 228 - Some studies have been conducted to assess the PMI.  This sentence appears incomplete.  Did you mean the study of glucose?

Discussion – The section is a single long paragraph.  Suggest breaking it up into multiple paragraphs.

Author Response

Dear Reviewer,

We appreciate the time and effort that you have dedicated to providing your valuable feedback on our manuscript entitled “Biochemical and Histological Assessment of post-mortem Changes to the Eyes of domestic pigs: A preliminary study” by Magdalena Palić, DVM, University Specialist in Veterinary Pathology; Associate Professor Ivan-Conrado Šoštarić Zuckermann DVM, Ph.D., DECVP; Full Professor Petar Džaja, DVM, Ph.D.; Blanka Beer Ljubić, Ph.D. and Full Professor Krešimir Severin, DVM, Ph.D. We agree with all your comments and thank you for your suggestions. Regarding the remarks on the quality of the English, we note that the manuscript was given for translation and review by an official Translator and Reviewer for English Language. The translator's CV is attached.

The corresponding author for negotiations concerning the manuscript is

Prof. Krešimir Severin, DVM, PhD

Department of Forensic and State Veterinary Medicine

Faculty of Veterinary Medicine, University of Zagreb

Ul. Vjekoslava Heinzela 55, Zagreb, Croatia

Email: severin@vef.unizg.hr

Kind regards,

Magdalena Palić, DVM, University Specialist in Veterinary Pathology

Department of Forensic and State Veterinary Medicine

Faculty of Veterinary Medicine, University of Zagreb

Ul. Vjekoslava Heinzela 55, Zagreb, Croatia

Reviewer 2 Report

Comments and Suggestions for Authors

 Review of : Biochemical and histological assessment of post-mortem 2 changes to the eyes of domestic pigs: A preliminary study

Title: ‘postmortem’ in the English language is one word. If latin is used, it is 2 words and italicized (post mortem). Please correct throughout the entire paper.

Please refer to the journals recommendations as to what tense to use. This paper is written mostly in present tense and most journals require past tense. Please consult the editor. 

Line11: From a veterinary forensics point of view, an accurate estimation of the postmortem interval is important when considering the worldwide increases in violent death of domestic and wild animals.

Line 17: the postmortem interval is the time since the death of an animal and its discovery.

Line 18 from a veterinary forensic standpoint, an accurate estimation of the PMI is of particular importance when estimating the postmortem interval especially with the observed increase in violent death……

Line 34: the postmortem interval is the time between the death of an animal…

Line 35 estimating the PMI is an important aspect of forensic investigations, therefore particular attention should be paid to practical as well as scientific research.

Line 78: 2.1. Animal sampling

Line 86: 2.2. Tissue sampling

Line 109: 2.5. Fixation and histological specimen preparation - categorical analysis

Line 111: considering the easy access to eyeballs off slaughtered pigs, using three animals per time points is a statistical challenge. The statistical power and significance could have been improved using at least 5 to 10 animals per time point.

Line 196: the images have no arrows.

Line 209-219: this sounds more like information of an introduction. This could be repeated but summarized into one sentence in the discussion section.

Line 221: In a study conducted on donkeys, a slight increase in sodium and chloride levels were recorded in the early PMI, which was not observed in our study.

Line 223: According to a previous study conducted exclusively on humans, postmortem concentrations of glucose in the vitreous were able to determine the cause of death, or otherwise indicates a previous lifelong hyperglycemia, or confirming diabetes and diabetic ketoacidosis as well as other causes of death, such as asphyxia…..

Line 229: In the present study, the concentrations of glucose were significantly decreasing after death in each animal.

Line 230: the mechanism of these changes in biochemical parameters can be explained in several ways.

Line 234: …ceases….

Line 236: The decrease in concentration of glucose……

Line 240: With regards to the nervous system,……

Line 248: In the present study, we aimed to reduce contamination to ….

Line 262: In the light of this, potassium levels higher than the reference values could be explained by the perimortem stressor and lethal agony, that lead to changes of the blood pH including extracellular/ metabolic acidosis.

Line 316: …. we were encouraged by previous studies in both…..

Comments on the Quality of English Language

As much as I understood what the authors meant to say, the overall quality of the English language should be addressed and improved. Most of my corrections are related to this.

Author Response

Dear Reviewer,

We appreciate the time and effort that you have dedicated to providing your valuable feedback on our manuscript entitled “Biochemical and Histological Assessment of post-mortem Changes to the Eyes of domestic pigs: A preliminary study” by Magdalena Palić, DVM, University Specialist in Veterinary Pathology; Associate Professor Ivan-Conrado Šoštarić Zuckermann DVM, Ph.D., DECVP; Full Professor Petar Džaja, DVM, Ph.D.; Blanka Beer Ljubić, Ph.D. and Full Professor Krešimir Severin, DVM, Ph.D. We agree with all your comments and thank you for your suggestions. Regarding the remarks on the quality of the English, we note that the manuscript was given for translation and review by an official Translator and Reviewer for English Language. The translator's CV is attached. Given the note regarding the number of samples, we have to emphasize that this is a preliminary study. We went with the smallest possible number to determine the normality of the distribution. Despite the number, normal distribution was maintained, and a larger number was unnecessary. Given the results obtained, the number of samples will increase in future planned studies. The aim was to see the distribution in as many time intervals as possible. The number of samples is determined by the research plan and this number cannot be increased in this study.

The corresponding author for negotiations concerning the manuscript is

Prof. Krešimir Severin, DVM, PhD

Department of Forensic and State Veterinary Medicine

Faculty of Veterinary Medicine, University of Zagreb

Ul. Vjekoslava Heinzela 55, Zagreb, Croatia

Email: severin@vef.unizg.hr

Kind regards,

Magdalena Palić, DVM, University Specialist in Veterinary Pathology

Department of Forensic and State Veterinary Medicine

Faculty of Veterinary Medicine, University of Zagreb

Ul. Vjekoslava Heinzela 55, Zagreb, Croatia

Reviewer 3 Report

Comments and Suggestions for Authors

The paper reports a preliminary study carried out on domestic pig eyes using biochemistry and histology to detect chemical and morphologic changes correlated to the postmortem interval (PMI).

The study design is well-developed and sound, and the results appear attractive, as they can impact the forensic sciences community (e.g., in the literature, I found no report of histological changes of the retina after death).

However, some revisions are mandatory. Please see below.

·      Regarding statistical analysis, I could not determine if the authors carried out the normality test before using mean values instead of median values or vice versa. I recommend adding a short statement with the results about this aspect (for example, applying the Kolmogorov-Smirnov test to assess the normality distribution).

·      As a limitation, the Author stored the eyes at 4°C during the experiment. I recommend adding a statement that, in a real environment, higher temperatures could affect this study’s results. 

·      Authors should warn readers that the proposed techniques may only sometimes be used. For example, in the case of an extended stay in the water, the electrolyte content could undergo modifications (please add reference: A.D. Cala, R. Vilain, R. Tse. (2013). Elevated postmortem vitreous sodium and chloride levels distinguish saltwater drowning (SWD) deaths from immersion deaths not related to drowning but recovered from saltwater (DNRD), Am. J. Forensic 310 Med. Pathol. 34, 133–138) and in some environments, the eye may not be usable due to the scavenger activity (Please add references: Pascali J. P., Viel G., Cecchetto G., Pigaiani, N. Vanin, S., Montisci M., Fais P. (2020). The red swamp crayfish Procambarus clarkii (the Louisiana Crayfish) as a particular scavenger on a human corpse. Journal of Forensic Sciences, 65(1), 323 -326.; O'Brien R. C., Forbes S. L., Meyer J., & Dadour I. R. (2007). A preliminary investigation into the scavenging activity on pig carcasses in Western Australia. Forensic science, medicine, and pathology, 3, 194-199.)

Comments on the Quality of English Language

English grammar and expression require correction before publication can be considered. Authors should get a professional language service to help with English expression and grammar, as writing in another language can be challenging. Alternatively, a native English speaker could revise the manuscript.

Author Response

(The authors gave the same response as above.)
